# Combined and Sequential Treatment with Deep Brain Stimulation and Continuous Intrajejunal Levodopa Infusion for Parkinson’s Disease

**DOI:** 10.3390/jpm11060547

**Published:** 2021-06-12

**Authors:** Daniël van Poppelen, Annelie N.M. Tromp, Rob M.A. de Bie, Joke M. Dijk

**Affiliations:** Department of Neurology, Amsterdam Neuroscience, Amsterdam UMC, University of Amsterdam, Meibergdreef 9, 1105 AZ Amsterdam, The Netherlands; d.vanpoppelen@amsterdamumc.nl (D.v.P.); annelie_tromp@live.nl (A.N.M.T.); r.m.debie@amsterdamumc.nl (R.M.A.d.B.)

**Keywords:** deep brain stimulation, levodopa/carbidopa enteral infusion, levodopa, Parkinson’s disease

## Abstract

(1) Background: Deep brain stimulation (DBS) and continuous intrajejunal levodopa infusion (CLI) are efficacious treatments of medication related motor response fluctuations in advanced Parkinson’s disease (PD). Literature regarding the use of both advanced treatments within one patient is scarce. (2) Methods: We present a retrospective single center case series and a review of the literature. Patients with PD who were treated with both DBS and CLI in our tertiary referral center between 2005 and 2020 were identified and medical records were assessed. Additionally, literature on patients treated with both therapies was systematically searched for in Medline and Embase. (3) Results: Nineteen patients were included. Medication related motor response fluctuations were a major indication for the second therapy in all but one. Of nine patients initially treated with DBS, five reported improvement with CLI. Seven of ten patients initially treated with CLI experienced benefits from DBS. The systematic literature search resulted in fifteen previous publications comprising 66 patients. Of the 59 patients, for whom the effect of the second treatment was known, 57 improved. (4) Conclusions: PD patients, who have persisting medication related motor response fluctuations, despite DBS or CLI treatment, may benefit from an additional or alternative treatment with either CLI or DBS.

## 1. Introduction

Both continuous deep brain stimulation (DBS) and intrajejunal levodopa infusion (CLI) are established treatments for advanced Parkinson’s disease (PD). Randomized controlled trials have shown that both interventions improve motor response fluctuations, the severity of symptoms during the off-drug phase, dyskinesias, disability, and quality of life [1,2]. With direct head-to-head comparisons lacking, there is no clear superior treatment and choice between treatments is based on various factors [3,4,5]. DBS has been available in the Netherlands since 1994 and CLI since 2004. In the Netherlands, both therapies are registered for advanced PD and are unconditionally reimbursed by all Dutch health insurers, so treatment is selected at the discretion of patients and physicians. This results in wide experience with these therapies. However, there is also a broad variation in practice regarding the advanced therapies; there are centers mainly providing DBS for advanced PD, while others only provide CLI or continuous subcutaneous apomorphine infusion (CAI) [6,7].

For patients with an unsatisfactory response to an advanced treatment, either initially or later, one may consider switching to, or adding a second advanced treatment. There is little experience and knowledge about the effects hereof. For the last fifteen years, patients have been referred to our tertiary movement disorders clinic and DBS center in the Netherlands for DBS after CLI treatment, whilst CLI treatment is also initiated in some patients that were initially treated with DBS. In this study, the clinical data of these patients are systematically assessed to evaluate if an additional advanced treatment is meaningful. Furthermore, a systematic review of published cases is provided.

## 2. Materials and Methods

A single center retrospective study was conducted at Amsterdam UMC in the Netherlands, a center experienced in both DBS and CLI treatment. Patients with Parkinson’s disease, who were simultaneously or sequentially treated with DBS and CLI between May 2005 and September 2020, were identified; our center’s CLI registry was screened for patients who had previously, simultaneously or subsequently been treated with DBS. Similarly, the DBS registry was screened for patients also treated with CLI at any time in their disease course. The retrieved cases were verified for completeness with the PD neurologists and PD specialized nurses, and possible additional identified cases were added. Patients participating in an ongoing randomized controlled trial on DBS and CLI treatment for advanced PD were excluded [8]. From the patient records, the following baseline characteristics were recorded: sex, age of onset of PD, other advanced treatment previous to DBS and CLI (e.g., thalamotomy, CAI), age at initiation of initial advanced treatment and time between start of the advanced therapies. The Unified Parkinson’s Disease Rating Scale (UPDRS) motor scores in the standardized off-drug phase (defined as at least 12 h without dopaminergic medication) and on-drug phase (with dopaminergic medication) before initiation of DBS were registered. When obtainable, oral dopaminergic treatment before the initial, and before and after the second advanced treatment were recalculated as levodopa equivalent daily dose (LEDD) [9]. Patient records were assessed for reporting on the initial efficacy of primary therapy, preferably approximately six months after initiation. If first treated elsewhere, patients’ referral letters to our center were used. To compare groups, the initial effect was dichotomized as beneficial in case of some or substantial improvement and as not beneficial when response was reported as no improvement or deterioration. One or more indications for initiating the second therapy reported in the patient records were first copied verbatim, and subsequently categorized in: motor response fluctuations, dyskinesias, dystonia, gait impairment, balance problems, speech impairment, hallucinations, cognitive decline, and device specific adverse effects by A.N.M.T. and D.v.P. For assessment of efficacy of the second advanced therapy, information hereon in the patient records approximately six months after initiation of the therapy was extracted. Descriptions of change of the general level of functioning were classified into one of the categories of a five-point scale (substantial deterioration, some deterioration, no change, some improvement, and substantial improvement) independently for each patient by two separate investigators (A.N.M.T., D.v.P.), when possible. If the outcomes contrasted, a third assessor (J.M.D.) was asked to decide on the classification. To compare groups, the effect of the second advanced therapy was further dichotomized as beneficial in case of some or substantial improvement, the other categories were qualified as not beneficial. The groups experiencing a beneficial effect were compared to the group without a beneficial effect using the Mann–Whitney U test or Fishers’ exact test, where appropriate. A *p*-value of less than 0.05 was deemed statistically significant. IBM SPSS Statistics v.26.0, IBM, Armonk, NY, software was used for the statistical analyses.

A systematic review of literature on patients subsequently or simultaneously treated with DBS and CLI was conducted through searches in Medline and Embase entailing all English abstracts, case reports, clinical trials, reviews and conference papers published until April 2021, using the following terms and available synonyms: Parkinson’s disease, continuous intrajejunal levodopa infusion and deep brain stimulation. See Appendix A for full search criteria. Titles and abstracts were screened for relevance by D.v.P. Possibly relevant papers were explored full-text and data from full texts were extracted with a standardized form. A 10% random sample of titles and abstracts was reassessed by J.M.D. Data from included papers were systematically subtracted, assessing the number of cases, sex, age at PD diagnosis, age at initial treatment, interval between treatments, DBS target, continuation of initial therapy, UPDRS pre-DBS and LEDD. If the latter was unavailable, it was recalculated [9]. Reported indications for initiation of the second therapy were recorded, more than one indication per patient was allowed. Improvement after the second advanced treatment, in general and specific for the major indication, were subtracted if possible.

Data of current and previously published cases were combined to assess overall prevalence figures or weighted means for the DBS target, sex, age at PD diagnosis, age at initial treatment, interval between treatments, beneficial effect of initial treatment, indication for the second advanced treatment and improvement hereafter.

## 3. Results

### 3.1. Retrospective Study

#### 3.1.1. Initial Treatment with DBS

Nine patients initially treated with DBS of the subthalamic nucleus (STN) with simultaneous or sequential CLI-treatment between May 2005 and September 2020 were identified (Table A1 in Appendix B). In the same fifteen-year period, a total of 663 PD patients were treated with DBS in our center. Of two patients treated with both DBS and CLI, only limited data and baseline characteristics were available since the therapies were initiated in other centers, one patient had been treated elsewhere with DBS as an initial treatment. Four of nine patients were male. Mean age at PD diagnosis was 43 years (range 33–52) and mean age at initiation of DBS was 53 years (range 42–65). Three patients had previously been treated with unilateral pallidotomy, one patient with unilateral thalamotomy, and one patient with continuous subcutaneous apomorphine infusion. In five of eight patients, good or excellent initial response to DBS was reported; data of one patient were missing. Reasons for CLI were (more than one possible): motor response fluctuations in all patients, impairment of gait in four, and balance impairment in one patient as well as an infection of the DBS system in one patient. Side effects and complications of DBS were reported in six out of nine patients, these included speech and balance impairments, and infection of the DBS system. The mean interval between the initiation of DBS and CLI was 57 months (range 41–88 months).

Patient records of five patients indicated improvement in general level of functioning approximately six months after CLI initiation; three of these patients were considered to have had a substantial improvement. In three patients, no change of functioning was recorded after additional treatment with CLI. In all patients, the STN stimulation was continued after CLI initiation; in two unilaterally because one electrode had been removed prior to CLI due to an infection. In one patient, CLI treatment was discontinued because of pain and multiple tube luxations.

Mean LEDD prior to DBS treatment was 838 mg/day (standard deviation (SD) 638, *n* = 7), mean LEDD prior CLI was 1164 mg/day (SD 470, *n* = 8) and with both therapies the mean LEDD was 1870 mg/day (SD 1049, *n* = 7).

#### 3.1.2. Initial Treatment with CLI

Ten patients, initially treated with CLI with simultaneous or sequential DBS-treatment between May 2005 and September 2020, were identified (Table A2). All patients were male. Mean age at PD diagnosis was 50 years (range 37–64) and mean age at initiation of CLI was 58 years (range 47–74). All patients had been treated with CLI in other clinics. The interval between the initiation of the first and second advanced treatment was 51 months (range 18–102). In eight patients, a good or excellent initial effect of CLI was reported. Indications for subsequent DBS treatment were (more than one possible): motor response fluctuations in nine patients, gait impairment in four, off-phase dystonia in one, painful polyneuropathy in one, inconvenience due to the CLI pump system in one and anxiety in the off phase in one patient. In six patients, CLI treatment was continued after initiation of DBS.

In seven patients, improvement in the general level of functioning approximately six months after initiation of the second therapy DBS was indicated; in two of them, this was considered a substantial improvement. In one patient, functioning had not changed after additional DBS treatment and for one patient transient slight worsening of the general level of functioning at six months was described, which had improved several months later. The data on efficacy from one patient were not available.

The mean LEDD prior to start with initial treatment of CLI was 1927 mg/day (SD 407, *n* = 6), mean LEDD prior to DBS was 2448 mg/day (SD 647, *n* = 10) and after initiation of both therapies the mean LEDD was 1418 mg/day (SD 441, *n* = 6) for patients continuing both therapies and 597 mg/day (SD 300, *n* = 4) for patients in whom CLI was discontinued.

Patients, both with initial DBS and CLI treatment, who did improve from the second therapy (*n* = 12) were compared to the patients who did not (*n* = 5); (Appendix A). There were no significant differences between these small groups when comparing sex, initial treatment, effect of initial treatment, mean age at PD diagnosis, mean age at initiation of the initial treatment, mean interval between treatments and mean final LEDD.

### 3.2. Results from the Systematic Review

Out of 574 papers, eventually eight full-text studies and seven conference abstracts were included (Figure 1). A blinded reassessment of title and abstract of 58 (10%) papers resulted in the selection of one additional paper for full-text evaluation that was not included ultimately. In the identified publications, patients with PD first treated with DBS and subsequently with CLI were described in 12 studies (total of 61 patients) [10,11,12,13,14,15,16,17,18,19,20,21]. In one published case-report, a single patient was described and in three conference abstracts a total of five patients first treated with CLI who were treated DBS thereafter were described [21,22,23,24].

The data of the previously published patients are summarized in Table A3 (initial treatment with DBS) and Table A4 (initial treatment with CLI). The 61 patients initially treated with DBS had a mean age at PD diagnosis of 48 years (range 30–70 years, *n* = 48) and a mean age at DBS of 62 years (range 40–77 years, *n* = 29). The mean interval between initiation of DBS and CLI was 7 years (range 2–18 years, *n* = 47). In 27 out of 32 patients a good initial effect of the first treatment DBS was reported (22 out of 24 patients treated with the STN DBS and five out of six patients treated with GPi DBS). In two patients, there was a modest effect of the first treatment of DBS and in three patients DBS had not led to any improvement; in 29 patients, the initial effect had not been described. Unpredictable motor fluctuations were an indication for the second advanced treatment in 40 out of the 54 patients for whom this could be determined (74%). Other indications were cognitive impairment (3/54), dystonia (6/54), increased off time (2/54), pain (2/54) and freezing of gait (8/54). These latter patients were reported in two prospective series of DBS patients in whom additional CLI was initiated specifically because of freezing of gait [20] and related walking difficulties [17].

Based on mostly retrospective data, 54 out of 56 (96%) DBS patients improved after subsequent CLI treatment. In the prospectively studied patients, in whom freezing of gait and related difficulties were the major indication for subsequent CLI, improvement hereof was reported in seven out of eight patients.

In the largest published cohort, 19 cases first treated with the bilateral STN stimulation were described. Although improvement was not reported on an individual level, after additional initiation of CLI, mean motor examination scores had improved. Fourteen patients had continued the CLI treatment, of whom five patients had discontinued DBS after battery depletion without clinical deterioration [12]. In combined data of the other studies, 21 out of 23 (91%) patients had continued their initial DBS treatment.

Five previously published cases initially treated with CLI with subsequent DBS were found. For two of these, only the indication for second treatment and DBS target could be retrieved [21]. One group reported two separate cases of men treated with DBS (one bilateral STN, one bilateral GPi) for recurrence of unpredictable response fluctuations and dyskinesias after initial treatment with CLI, which had initially been successful in one patient [22,23]. In an additional case report, a woman was described who was diagnosed with PD at 29 years and treated with DBS after CLI for dyskinesias and painful dystonia [24]. For the three reported cases in whom this was known, all improved after DBS and none continued CLI treatment.

### 3.3. Pooled Results

Combining the data of 19 cases treated in our center and from all retrieved previously published cases, data of 70 patients firstly treated with DBS and subsequently with CLI and of 15 patients with CLI as first and DBS as subsequent treatment are available, adding up to a total of 85 patients. See Table 1 for pooled results. It was not possible to assess improvement for separate indications for second advanced treatment, as multiple indications per patient were possible and the effect of treatment for separate indications was not reported consistently.

## 4. Discussion

Over a fifteen-year period, nineteen patients were treated with both DBS and CLI in our tertiary movement disorders clinic, either simultaneous or sequentially. The data of our patients, combined with previously published cases, comprise a total of 85 patients. Results suggest a beneficial effect of CLI therapy after initial treatment with DBS in 60 out of 64 patients (94%). Moreover, in 10 of the 12 patients initially treated with CLI (83%), improvement was reported after subsequent DBS therapy. Even though in most patients a good initial effect of the first treatment was reported, the major indications for additional advanced treatment were recurrence of medication induced motor response fluctuations and bothersome dyskinesias during treatment with the solitary initial therapy.

Both DBS and CLI are known to remain effective over time. Follow-up studies have shown a favorable effect on motor symptoms up to five years after initiation of CLI, and up to ten years after initiation of DBS [25,26]. In the patients we have described nonetheless, recurrence of unpredictable motor fluctuations during the first advanced treatment seemed a main indication for turning to a second advanced therapy. Several factors may have contributed to this remarkable observation. First, patients with a suboptimal long-term effect of an advanced therapy may either not have been included in the studies reporting long-term results or may not be noticed in the reported group means. Second, side effects of the first advanced treatment may have hampered optimization of treatment, and, thus, may have led to persisting motor fluctuations. Third, the noticeable young age on initiation of the first advanced therapy in a large proportion of patients may have made them more vulnerable for recurrence of motor fluctuations due to a longer disease course after initiation of the primary therapy; at the same time, they then still were young enough to be eligible for a second advanced therapy. Moreover, the relatively young patients may have had higher expectations of the advanced therapy than elderly patients due to demanding activities of daily life and, therefore, may have been more dissatisfied with the effect of the first advanced treatment; this was not shown in a study on satisfaction with DBS treatment though [27]. Finally, unilateral DBS treatment in seven of the described patients may have led to persisting response fluctuations due to a disbalance in the requirements of dopaminergic medication between the hemisphere with and without DBS (i.e., higher dopaminergic requirements in the hemisphere not treated with DBS may have limited optimization of DBS settings contralaterally, leading to either persisting response fluctuations or dyskinesias).

Barring recurrence of motor fluctuations, other indications have led to initiation of the second advanced therapy. Troublesome complications of one of the therapies seemed a reason to switch to another therapy in several patients. Reported examples of complications of the first therapy were technical issues with CLI systems, infection in both CLI and DBS and other side effects such as speech and balance impairment after DBS and polyneuropathy due to CLI treatment. Furthermore, a relatively frequent indication for CLI in patients already treated with DBS was freezing of gate and related walking difficulties; in two observational studies, improvement of walking was reported in most patients after CLI [17,20].

The STN was the most common target in patients initially treated with DBS (61/69). This was likely due to the predominant selection of the STN as a target for DBS in PD in current practice. It could not be ruled out, however, that additional treatment with CLI was reported to a lesser extent in patients treated with GPi DBS, because this was necessary relatively less frequently.

Even though both advanced therapies are known to be relatively expensive, in many patients the first advanced treatment was continued. A study estimated a mean cumulative 5-year cost of EUR 89,477 for DBS and EUR 234,643 for CLI treatment alone [28]. Of the three device-aided therapies in PD, DBS was considered a cost-effective therapy; this was not so clear for CLI and CAI [29]. Although our study did not cover costs, these should be taken into account when introduction of a second advanced treatment is considered.

This review is, to our knowledge, the first overview of systematically collected data on the effect of a second advanced treatment in PD and includes a considerable number of additional cases from our own center. This provides necessary knowledge to guide treatment. That all our own cases over an extensive period were added, is also relevant as there is no publication bias in this series of patients and because we report on the first significant series of patients firstly treated with CLI.

Some limitations of the case series and systematic review must be mentioned as well. Even though not likely, it cannot be ruled out that patients treated with both DBS and CLI in our center were missed. Moreover, information in our case series was retrospectively obtained from clinical files and was sometimes open to interpretation; to minimize potential resulting bias, the assessment was conducted by two investigators. Furthermore, the effect of the second treatment was evaluated at six months, since potential effects over a longer retrospective period may have been blurred by other influences on general health and wellbeing. Still, a longer follow-up duration could have been of interest as well. Additionally, standardized test results regarding CLI treatment were scarce as in most patients CLI was initiated elsewhere.

Regarding the review, as most included studies did not report on all consecutive patients treated with both treatments in a certain period, and since in the large majority of published patients a beneficial effect of the second treatment was reported, publication bias may have played; then, possibly overestimating the success of a second advanced treatment. Nonetheless, our relatively large single center cohort, including all consecutive patients, also showed a positive effect. In a few patients, additional treatment with CAI was mentioned. As the focus of this study was on DBS and CLI, treatment with CAI was not systematically assessed. Therefore, it cannot be ruled out that other patients, either in our own cohort or previously reported, were additionally treated with CAI.

Based on the results of our case series and systematic review showing a beneficial effect of a second advanced treatment in most patients, a second advanced treatment can be considered in eligible patients with recurrence of medication induced motor fluctuations. Specifically, more data support that patients initially treated with DBS may benefit from additional CLI. Furthermore, results of eight published cases show that additional CLI in DBS patients may improve freezing of gait. Of course, whether initiation of a second advanced therapy in PD can be installed also depends on local regulations, costs and availability of device-aided therapies. To additionally explore possible benefits, risks and costs from adding a second advanced treatment, larger, prospective studies are needed.

## 5. Conclusions

In our case series and literature review on patients treated with both DBS and CLI, the majority of patients seemed to benefit from the second advanced therapy. Most evidence was available for patients initially treated with DBS and secondly with CLI. The main indications for a second therapy were development or recurrence of medication induced motor response fluctuations and dyskinesias during treatment with the initial therapy. Based on the results in this study, adding a second advanced therapy should be considered in eligible patients suffering from motor response fluctuations or dyskinesias after treatment with DBS or CLI if permitted by local availability and regulations.

## Figures and Tables

**Figure 1 jpm-11-00547-f001:**
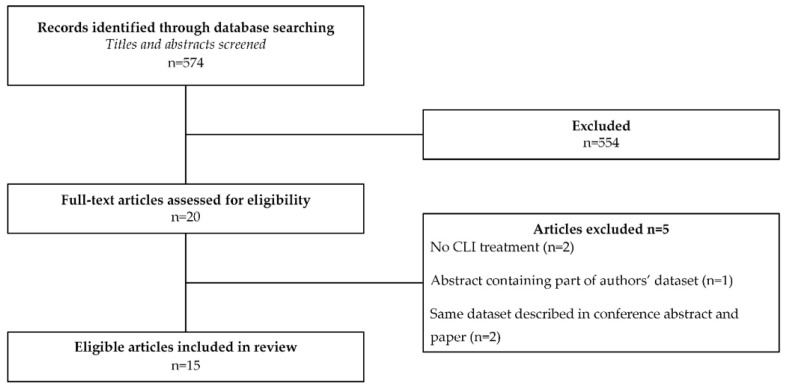
Flow chart: Selection of Studies.

**Table 1 jpm-11-00547-t001:** Pooled results from currently presented and previously reported cases.

	Initial Treatment DBS (*n* = 70)	Initial Treatment CLI (*n* = 15)
Sex F (*n* out of total *n*; (%))	27/61 (44%)	1/12 (8%)
Age at PD diagnosis in years; WM (range, *n*)	47 (30–70, *n* = 57)	49 (29–64, *n* = 13)
Age at initiation of initial treatment in years; WM (range, *n*)	60 (40–77, *n* = 38)	57 (47–67, *n* = 13)
Beneficial effect of initial treatment (*n* out of total *n*; (%))	32/40 (80%)	10/12 (83%)
Major indication for 2nd treatment (more than one possible)		
*MFD (n out of total n; (%))*	45/59 (76%)	14/15 (93%)
*GI/FOG (n out of total n; (%))*	14/59 (24%)	2/15 (13%)
Interval between treatment in years; WM (range, *n*)	6.9 (0–18, *n* = 57)	4.2 (2–9, *n* = 13)
Beneficial effect of second treatment (*n* out of total *n*; (%))	60/64 (94%)	10/12 (83%)
DBS target		
*STN*	87%	87%
*GPi*	10%	13%
*PPN*	3%	0%

CLI: Continuous levodopa/carbidopa infusion; DBS: deep brain stimulation; GI: gait impairment; F: female; FOG: freezing of gait; GPi: globus pallidus interna; M: male; MFD: unpredictable motor fluctuations and dyskinesias; PD: Parkinson’s disease; PPN: pedunculopontine nucleus; STN: subthalamic nucleus; WM: weighted mean.

## Data Availability

Available data is incorporated in this paper.

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
