# Peer review of "Combined and Sequential Treatment with Deep Brain Stimulation and Continuous Intrajejunal Levodopa Infusion for Parkinson’s Disease"

_jpm, 2021, doi:10.3390/jpm11060547_

Round 1

Reviewer 1 Report

It would be useful to know if these patients had also previously apomophine pump.

Author Response

We would like to thank the reviewer for reviewing our manuscript and value the suggestion, which we will address below.

Response: One patient in our study group has previously been treated with continuous apomorphine infusion (CAI) (see line 121). With respect to the previous reported cases, to our knowledge, only one patient in the study by Sanchez et al. (2019), has been treated with CAI after trying brain stimulation (DBS) and continuous levodopa infusion (CLI). This has been added in table A3 as a footnote in the revised manuscript. Furthermore,  Spanaki et al. published a series of eight patients, of which five patients initially treated with DBS received CLI, one initially treated with DBS was treated with CAI and two patients initially treated with CLI were treated with DBS as second advanced therapy. No data on an individual level were available. This is added to table A3 as a footnote. 

We added a paragraph to the manuscript to discuss that, as the focus of our manuscript is on DBS and CLI, treatment with continuous apomorphine has not been systematically assessed. Therefore, it cannot be ruled out that other patients, either in our own cohort or previously reported, have been additionally treated with CAI after treatment with DBS and CLI (see line 297-300).

Reviewer 2 Report

I have some minor suggestions: 

1) It should be useful to give pooled data of the authors study and literature review, including two additional tables A5-A6, in the results section (not simply mentioning these results in the discussion).

2) It should be useful to give data on the efficacy of the 2nd therapy after a follow-up period if this is possible.  

Author Response

We would like to thank the reviewer for reviewing our manuscript and value the suggestions, which we will address below.

Response to suggestion 1: A paragraph and an additional table describing pooled data of patients first treated with DBS and thereafter with CLI, and vice versa, has been added to the new version of the manuscript (table 1, line 211-218). The Materials & Methods sections were adjusted accordingly (line 98-108).

Response to suggestion 2: information regarding a longer follow-up period is certainly relevant and of interest, but we have not systematically evaluated the effect of the second therapy beyond six months. We have chosen this six-month follow-up duration, because we considered this a reasonable time point to evaluate the optimal effect of the additional treatment. Besides, after six months, all patients were still under our treatment, whilst thereafter a proportion of patients was referred to other clinics for continuation of treatment. Finally, we considered it likely that assessment of the effect the additional treatment retrospectively after a longer follow-up duration is more biased due to disease progression and potential changes in general health.  

Reviewer 3 Report

The manuscript entitled Combined and sequential treatment with deep brain stimulation and continuous intrajejunal levodopa infusion for Parkinson’s disease, by van Poppelen et al is a very interesting manuscript devoted to a poorly addressed aspect of the clinical practice, as what to do after a therapeutic failure. Nevertheless, I have some worries about the formal questions of the manuscript.

1.- Quantification is poorly addressed and authors are more devoted to qualitative description. There is not statistical analysis. Authors have not compared groups; therefore, they do not have described statistical methods. However, I think that some comparison can be done, e.g, between patients with good and bad response to a second therapy. Therefore, some methodological explanation should be deserved at Methods about statistics, more than the commercial packaged used (line 80). Along the manuscript there are several expressions too vague and poorly scientific, e.g “83% [patients] appeared to have improved” (line 194).

2.- The criteria for systematic review should be at least stated, although detail will be offered at supplement.

3.- In order to have an idea about the number of patients needing a second therapy, please indicate the number of patients treated by DBS/CLI at the same period.

4.- Why not use mensurable scales (obviously more objective than clinical perception) to quantify clinical results?. Moreover, it would be important to show if there are clinical changes along the evolution (maybe showing graphs).

Author Response

We would like to thank the reviewer for reviewing our manuscript and value the suggestions, which we will address below.

  1. Quantification is poorly addressed and authors are more devoted to qualitative description. There is not statistical analysis. Authors have not compared groups; therefore, they do not have described statistical methods. However, I think that some comparison can be done, e.g, between patients with good and bad response to a second therapy. Therefore, some methodological explanation should be deserved at Methods about statistics, more than the commercial packaged used (line 80). Along the manuscript there are several expressions too vague and poorly scientific, e.g “83% [patients] appeared to have improved” (line 194).

Response to suggestion 1: We agree with the reviewer that out results are mainly presented qualitatively. As mentioned in the manuscript, only retrospective data were available, limiting quantification. Nonetheless, the reviewer’s suggestion to compare groups of responders and non-responders to second treatment is very valuable. Following this, we added a paragraph and a supplementary table addressing these sub-groups of our patients to the result section (see line 149-151 and Supplement 2 – Table) and have edited the “Materials & Methods” section accordingly (See line 70-72, 83-88). This now also contains a more thorough description of applicable statistics (line 86-88).

The previously published cases contain little data to compare as nearly all described patients responded to second treatment; publication bias likely plays a role here. Therefore, the previously published cases were not included in the comparative analyses.

We have updated several expressions that might be interpreted as too vague (line 226, 228, 256, 262,263).

2.- The criteria for systematic review should be at least stated, although detail will be offered at supplement.

Response to suggestion 2: The search terms and criteria have been added to the Material and methods section in version 2 (line 92-94).

3.- In order to have an idea about the number of patients needing a second therapy, please indicate the number of patients treated by DBS/CLI at the same period.

Response to suggestion 3: in the described period, 633 patients have been treated with deep brain stimulation for Parkinson’s disease in our center. This figure is added to the new version of the manuscript, line 144. All patients initially treated with continuous levodopa infusion (CLI) received this treatment elsewhere, therefore a total number of patients treated with CLI in our center is of little significance.

4.- Why not use mensurable scales (obviously more objective than clinical perception) to quantify clinical results?. Moreover, it would be important to show if there are clinical changes along the evolution (maybe showing graphs).

Response to suggestion 4: We agree with the reviewer that mensurable or quantifiable scales are preferable to qualitative descriptions. For this reason, if available, the results of validated scales that were used in clinical practice such as the MDS-UPDRS and comparable units such as the levodopa equivalent daily dose are reported. For the effect of the second therapy, we had to rely on retrospective analysis of the patient files as there was no systematic assessment using validated scales in the period these patients were treated in our centre. Similarly, few quantitative scales besides (subsets of) the MDS-UPDRS were available in the cases in described literature. If available, usage of such scales varied greatly between the studies, hampering comparability (e.g. quantitative measures for walking for three patients (Kimber et al., 2019), frontal assessment battery and MMSE for three patients (Bautista et al. 2020), PDQ-39 for four other patients (Elkouzi et al.  2019))

Clinical changes along the evolution in time proved difficult to assess and unfortunately could not be added to the manuscript. In our patient series, we pragmatically have chosen the follow-up interval of 6 months, since all patients were still under treatment in our center at that time and we considered this as a reasonable time point to evaluate the effect of the second therapy after an optimization phase. The effect of the therapies after a longer treatment duration was not systematically assessed. This could have been of additional value, although we feel that this can be blurred by changes in general health due to other causes, when assessed retrospectively. As practically all previously published cases responded to second treatment irrespective of the treatment duration (publication bias may play a role here), a graph illustrating evaluation of clinical effect over time would not be of additional value. That 57 of 59 previously published cases responded has been added to the abstract for clarification. (Line 20)

Round 2

Reviewer 3 Report

The manuscript has been considerably improved. I have no more concerns.